# Prolonged Lifespan, Improved Perception, and Enhanced Host Defense of *Caenorhabditis elegans* by *Lactococcus cremoris* subsp. *cremoris*

Tomomi Komura,ᵃ* Asami Takemoto,ᵃ Hideki Kosaka,ᵇ Toshio Suzuki,ᵇ ⓘYoshikazu Nishikawaᶜ§

ᵃFaculty of Human Life and Environment, Nara Women's University, Nara, Japan
ᵇFujicco Co., Kobe, Hyogo, Japan
ᶜGraduate School of Human Life Science, Osaka City University, Osaka, Japan

**ABSTRACT** Lactic acid bacteria are beneficial to *Caenorhabditis elegans*; however, bacteria acting as probiotics in nematodes may not necessarily have probiotic functions in humans. *Lactococcus cremoris* subsp. *cremoris* reportedly has probiotic functions in humans. Therefore, we determined whether the strain FC could exert probiotic effects in *C. elegans* in terms of improving host defenses and extending life span. Live FC successfully extended the life span and enhanced host defense compared to *Escherichia coli* OP50 (OP50), a standard food source for *C. elegans*. The FC-fed worms were tolerant to *Salmonella enterica* subsp. *enterica* serovar Enteritidis or *Staphylococcus aureus* infection and had better survival than the OP50-fed control worms. Further, the chemotaxis index, an indicator of perception ability, was more stable and significantly higher in FC-fed worms than in the control worms. The increase in autofluorescence from advanced glycation end products (AGEs) with aging was also ameliorated in FC-fed worms. FC showed beneficial effects in *daf-16* and *pmk-1* mutants, but not in *skn-1* mutants. Since SKN-1 is the *C. elegans* ortholog of Nrf2, we measured the transcription of heme oxygenase-1 (HO-1), which is regulated by Nrf2, in murine macrophages and found that HO-1 mRNA expression was increased >5 times by inoculation with FC cells. Thus, FC could exert antisenescence effects via the SKN-1/Nrf2 pathway. This study showed for the first time that FC supported perceptive function and suppressed AGEs in nematodes as probiotic bacteria. Therefore, *C. elegans* can be an alternative model to screen for probiotic bacteria that can be used for antisenescence effects in humans.

**IMPORTANCE** Aging is one of our greatest challenges. The World Health Organization proposed that "active aging" might encourage people to continue to work according to their capacities and preferences as they grow old and would prevent or delay disabilities and chronic diseases that are costly to both individuals and the society, considering that disease prevention is more economical than treatment. Probiotic bacteria, such as lactobacilli, are live microorganisms that exert beneficial effects on human health when ingested in sufficient amounts and can promote longevity. The significance of this study is that it revealed the antisenescence and various beneficial effects of the representative probiotic bacterium *Lactococcus cremoris* subsp. *cremoris* strain FC exerted via the SKN-1/ Nrf2 pathway in the nematode *Caenorhabditis elegans*.

**KEYWORDS** *Caenorhabditis elegans*, *Lactococcus*, *Salmonella*, *Staphylococcus*, host resistance, immune senescence, longevity, probiotics

Address correspondence to Yoshikazu Nishikawa, y-nishikawa@tezuka-gu.ac.jp.

*Present address: Tomomi Komura, School of Human Science and Environment, Research Institute for Food and Nutritional Sciences, University of Hyogo, Himeji, Hyogo, Japan.

§Present address: Yoshikazu Nishikawa, Faculty of Human Sciences, Tezukayamagakuin University, Sakai, Osaka, Japan.

The authors declare no conflict of interest.

According to the World Health Organization, population aging is one of humanity's greatest triumphs, but it can also be our greatest challenge. Population aging has already been recognized in developing countries; by 2025, the number of people aged 60 and over will be approximately 840 million, representing 70% of the population of older people worldwide. The World Health Organization proposed "Active Aging: A Policy

Framework," which might encourage people to continue to work according to their capacities and preferences as they grow old. Considering that disease prevention is more economical than treatment, this would also prevent or delay disabilities and chronic diseases that are costly to both the individual and society (1).

Active aging can be accomplished by consuming healthy foods and nutrients. Based on the longevity of Bulgarians who consumed large quantities of yogurt, Metchnikoff hypothesized that lactobacilli are important for human health and longevity (2). This has accelerated research focusing on healthy bacteria. Probiotic bacteria are defined as live microorganisms that exert beneficial effects on human health when ingested in sufficient amounts (3). The influence of the microbiome on human health is becoming evident, and thus, the inclusion of probiotics, prebiotics, synbiotics, and biogenics in diet is expected to improve health (4, 5).

*Caenorhabditis elegans* is a small, free-living soil nematode (roundworm) that feeds on bacteria. This nematode has been extensively used as an experimental model for biological studies because of its simplicity, transparency, ease of cultivation, and suitability for genetic analysis (6). Furthermore, due to its short and reproducible life span, *C. elegans* is particularly suitable for aging studies (7). One century after Metchnikoff's study, we succeeded in showing that lactic acid bacteria (LAB) exert longevity effects in *C. elegans* (8). At present, the *C. elegans* model is being used worldwide to evaluate probiotic strains (9, 10), and the strains selected using this model have shown beneficial effects even in swine (11). However, it remains unclear whether the phenotypes observed in such simple creatures would be relevant and be extrapolated to humans (12).

The term LAB loosely includes a variety of bacteria that are Gram-positive, nonsporulating, and mainly produce lactic acid by fermentation. LAB have been used in various fermented foods since antiquity and are the most commonly used probiotic microorganisms. The spherical bacterium *Lactococcus cremoris* subsp. *cremoris* subsp. nov. (13), a LAB formerly known as *Lactococcus lactis* subsp. *cremoris*, is expected to have various physiological effects on humans consuming probiotic dairy foods (14). We have reported various beneficial effects of *L. cremoris* subsp. *cremoris* strain FC (FC) on colitis (15), immunomodulation (16), biopreservation, antimicrobial activities (17), dermatitis (18), and healthy defecation (19). Furthermore, other groups have reported the benefits of *L. lactis* for folate supplementation (20) and its inhibitory effects on food allergies (21), pollinosis (22), tumors (23), serum cholesterol (24), and hypertension (25). In addition, strain H61 was shown to suppress aging-associated symptoms, such as osteoporosis (26) and hearing loss (27), in senescence-accelerated mice.

In this study, we evaluated whether the human probiotic strain FC could improve the host defenses, retard the accumulation of advanced glycation end products (AGEs) and deterioration of perception ability, and prolong the life span of *C. elegans*. The life span and resistance to physical, chemical, and biological stressors were compared between *C. elegans* fed either FC or *Escherichia coli* OP50 (OP50), an international standard food for *C. elegans*. In addition, the mechanism of the effects of FC feeding was determined using *C. elegans* loss-of-function mutants for defense-related genes. Human macrophages were also inoculated with FC to investigate whether the bacteria could influence human innate immunity.

## RESULTS

**Amelioration of senescence.** To evaluate antisenescence effects of FC, the survival curve of FC-fed worms was compared to that of OP50-fed worms. Young adult worms were fed FC on the third day after hatching (nominal 0 days old). The average life span of the FC-fed nematodes was greater than that of OP50-fed nematodes (Fig. 1A). The life span of heat-killed OP50 (HKOP50)-fed *C. elegans* was longer than that of live OP50-fed *C. elegans* because of the loss of virulence (Fig. 1B). However, both live and heat-killed FC (HKFC) prolonged the life span of *C. elegans* compared to that of the HKOP50-fed nematodes. Thus, the longevity afforded by FC was not attributed to its nonpathogenic properties alone.

Locomotor ability was measured as an indicator of health span. The ratio of nematodes showing coordinated sinusoidal locomotion (class A) was higher in the FC-fed group than in the control nematodes fed OP50 (Fig. 1C): the survival curves of class A showed significant differences between the groups (Fig. 1D).

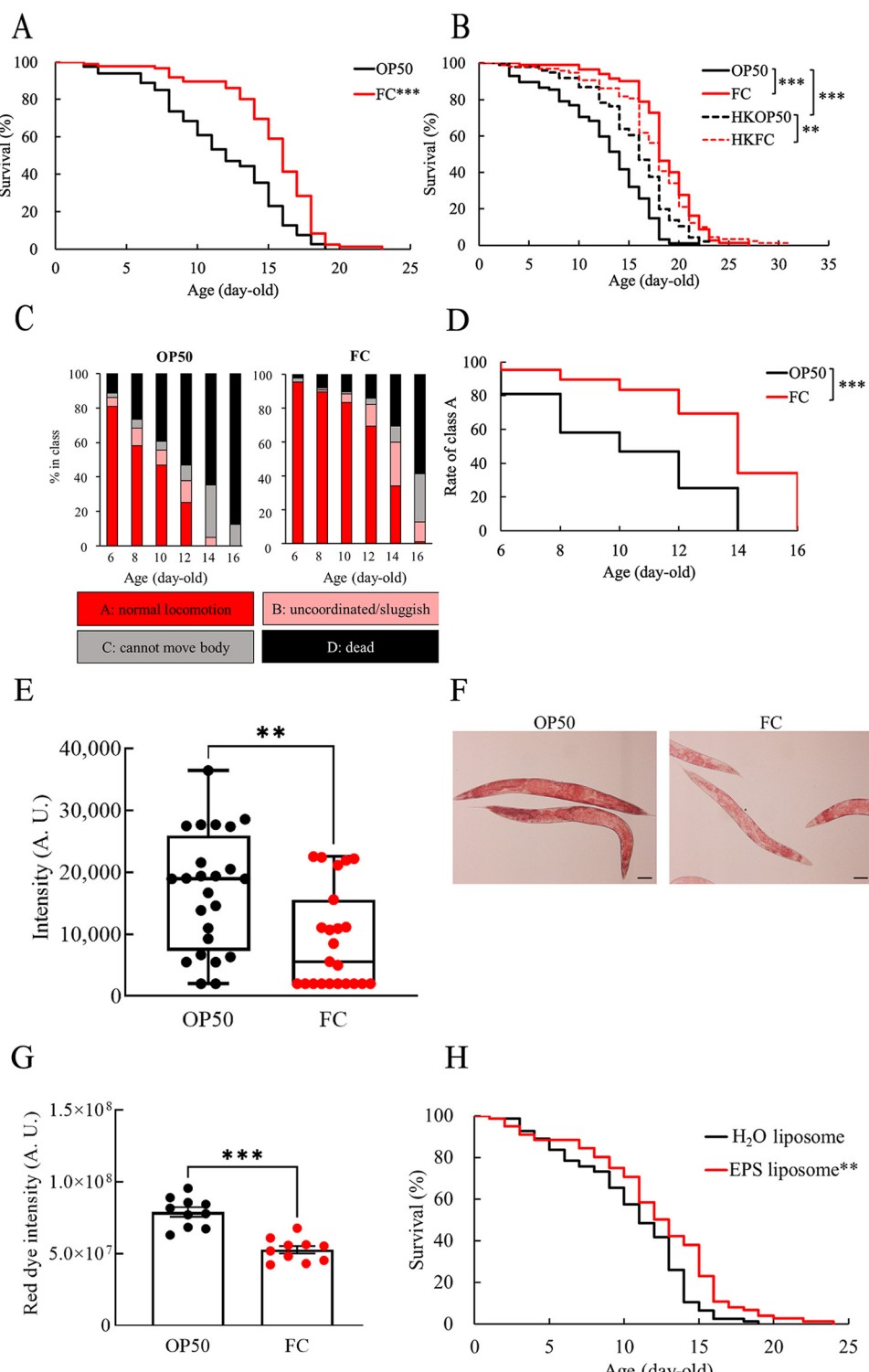

FIG 1 Antisenescence effects of *L. cremoris* strain FC (FC) on classical aging biomarkers. (A) Survival curves of FC-fed *C. elegans* compared with *E. coli* strain OP50 (OP50)-fed control worms. Young adult worms were defined as nominal 0 days old. Asterisks indicate significant differences (***, $P < 0.001$) compared to control worms using a log-rank test. OP50-fed worms, $n = 79$; FC-fed worms, $n = 85$; trial no. 1 (Table S1) is shown as a representative graph. (B) Survival curves of heat-killed bacteria-fed *C. elegans*. Young adult worms (0 days old) were placed on plates containing live bacteria (OP50, FC), heat-killed OP50 (HKOP50), or heat-killed FC (HKFC). Asterisks indicate significant differences (**, $P < 0.01$; ***, $P < 0.001$) from control worms fed live OP50 or HKOP50 using a log-rank test. Live OP50-fed worms, $n = 94$; live FC-fed worms, $n = 80$; HKOP50-fed worms, $n = 96$; HKFC-fed worms, $n = 91$; trial no. 2 (Table S1) is shown as a representative graph. (C) Locomotory activity of FC-fed *C. elegans*. Worms of each age were

Recently, we showed that blue autofluorescence is an indicator of aging and is higher in older *C. elegans* than in young ones. This was attributed to the AGEs of vitellogenins in the gonads (28). Since FC prolonged the life span of nematodes, we hypothesized that FC-fed nematodes might have less autofluorescence. Indeed, the intensity of blue autofluorescence in FC-fed *C. elegans* was weaker than that in control *C. elegans* (Fig. 1E). The effects of FC on the metabolism of *C. elegans* were also determined to assess how FC prolonged the life span of the nematodes. Oil red O staining showed that the FC-fed nematodes stored fewer lipids (Fig. 1F and G).

The production of exopolysaccharide (EPS) is a well-known characteristic of *L. cremoris* (29). As the longevity effect of heat-killed FC was somewhat lower than that of live FC, we attempted to determine whether the longevity effect was attributed to EPS purified from FC. Using our previously reported liposome method (30), liposomes, including acellular crude EPS, were administered to the nematodes with OP50. However, EPS failed to prolong the life span of the nematodes significantly (see Table S1 in the supplemental material); statistical significance was observed in only one of the three trials (Fig. 1H).

As FC extended the health span of *C. elegans*, we investigated whether the bacteria could slow the senescence of perception of *C. elegans* (Fig. 2A). The chemotaxis index (CI) refers to the percentage of the worms pointing to the attractant zone among the population served in test plates. The CI, which tended to decrease with aging (2 days old to 4 days old), was more stable and significantly higher in the nematodes fed FC (CI = 0.6) than in the control nematodes (CI = 0.2) at 4 days old (Fig. 2B).

We had shown that lactobacilli and bifidobacteria could increase the tolerance of nematodes to *Salmonella* infection (8). In this study, we investigated whether FC could enhance the host defense of *C. elegans* to *Salmonella enterica* subsp. *enterica* serovar Enteritidis (*S.* Enteritidis) and *Staphylococcus aureus* infection. FC-fed nematodes had better survival than the OP50-fed ones after infection with pathogens (Fig. 3A); this effect was unlikely to be attributed to the increased resistance, as no significant difference was noted in the number of pathogens recovered from both FC- and OP50-fed nematodes (Fig. 3B).

Integrity of the epithelial barrier is vital for host defense. Zhao et al. (31) reported that pathogens that are virulent to nematodes could invade oropharyngeal tissues. The OP50-fed old worms showed leakage of blue dye in tissues around the pharynx when they were 9 days old; however, the dye remained in the lumen, and the leak to the body cavity was not observed in FC-fed worms (Fig. 3C). When nematodes fed OP50 or FC for 4 days were infected with *S.* Enteritidis (Fig. 3D) or *S. aureus* (Fig. 3E), the worms (9 days old) fed OP50 before infection showed a higher leakage of dye into the body cavity than did the worms fed FC until infection; the barrier function of the epithelia in FC-fed nematodes was resistant to destruction by the pathogen (Fig. 3F).

FC-fed *C. elegans* was resilient against not only senescence and infectious diseases but also physical or chemical stresses. The survival of FC-fed nematodes was better than that of OP50-fed ones after exposure to either heat at 35°C or juglone (Fig. 3G); however, FC-fed nematodes could not live significantly longer than the OP50-fed ones after exposure to cupric chloride or paraquat (data not shown).

**FIG 1** Legend (Continued)

classified into four classes based on their locomotion: class A, robust, coordinated sinusoidal locomotion (red bars); class B, uncoordinated and/or sluggish movement (pink bars); class C, no forward or backward movement, but head movements or shuddering in response to prodding (gray bars); and class D, dead animals (black bars). The proportion of each class at the indicated time point is indicated. OP50-fed worms, $n = 90$; FC-fed worms, $n = 86$. (D) Health span curve of FC-fed *C. elegans* compared with that of OP50-fed control worms; percentages of worms in class A were traced over time. (E) Autofluorescence (e.g., 340 nm/430 nm) in 6-day-old worms grown with OP50 or FC was measured using the wrap-drop method. Asterisks indicate statistically significant differences (**, $P < 0.01$) from control worms fed OP50 determined by Mann-Whitney *U* test. OP50-fed worms, $n = 24$; FC-fed worms, $n = 23$. (F) Lipid accumulation in nematodes. Oil red O staining shows fat in 4-day-old worms grown with OP50 or FC. Scale bar indicates 100 $\mu$m. Magnification, ×100. (G) Red dye on 4-day-old worms was quantified based on the projection area of a worm's body. Error bars represent the standard error. Asterisks indicate statistically significant differences (***, $P < 0.001$) from control worms fed OP50 determined with Student's *t* test. (H) Survival curves of *C. elegans* fed the exopolysaccharide (EPS) extracted from FC culture. $H_2O$ liposome-fed worms, $n = 77$; EPS liposome-fed worms, $n = 74$; trial no. 3 (Table S1) is shown as a representative graph. Asterisks indicate significant differences (**, $P < 0.01$) from control worms fed $H_2O$ liposome determined with log-rank test.

A

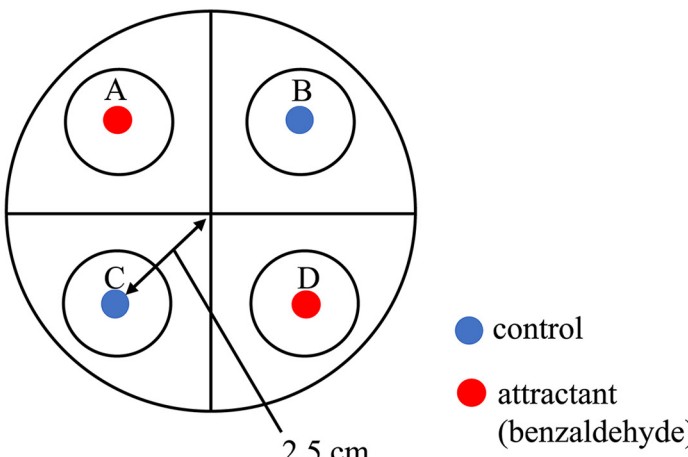

B

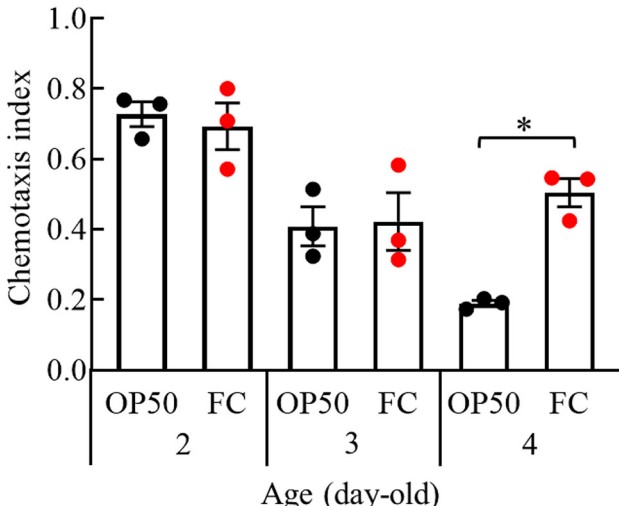

**FIG 2** Effects of FC on senescence of perception in *C. elegans*. (A) Schematic view of the chemotaxis assay. Assay plates were divided into two quadrants of attractants (benzaldehyde, regions A and D) and two of control (ethanol, regions B and C). (B) The chemotactic index (CI) of wild-type strain N2 fed OP50 or FC from day 0. The numbers of worms fed OP50 for 2, 3, and 4 days were 212, 210, and 217, respectively. The numbers of worms fed FC for 2, 3, and 4 days were 217, 207, and 218, respectively. Error bars represent the standard error. Asterisks indicate significant differences (*, $P < 0.05$; two-factor factorial ANOVA and Tukey-Kramer test).

**Mechanisms of antisenescence by FC.** Caloric restriction (dietary restriction in the case of *C. elegans*) is generally considered to extend the life span of various animals (32). The body size of the nematodes fed FC was slightly smaller than that of OP50-fed nematodes after the food source was changed at 0 days old (Fig. 4A); however, the brood size was the same as that of the control nematodes (Fig. 4B). In contrast, dietary restriction has been reportedly applied to worms previously sterilized with 5-fluorouracil because fertile worms escape from plates under dietary restriction (33, 34). Therefore, the longevity effects due to FC might not be attributed only to the so-called calorie restriction.

The mechanism underlying the antisenescence effects of FC, which prolonged the life span and enhanced the host defense of nematodes, was investigated by examining mutants lacking defense-related signaling pathways. DAF-16, the mammalian homolog of the FOXO

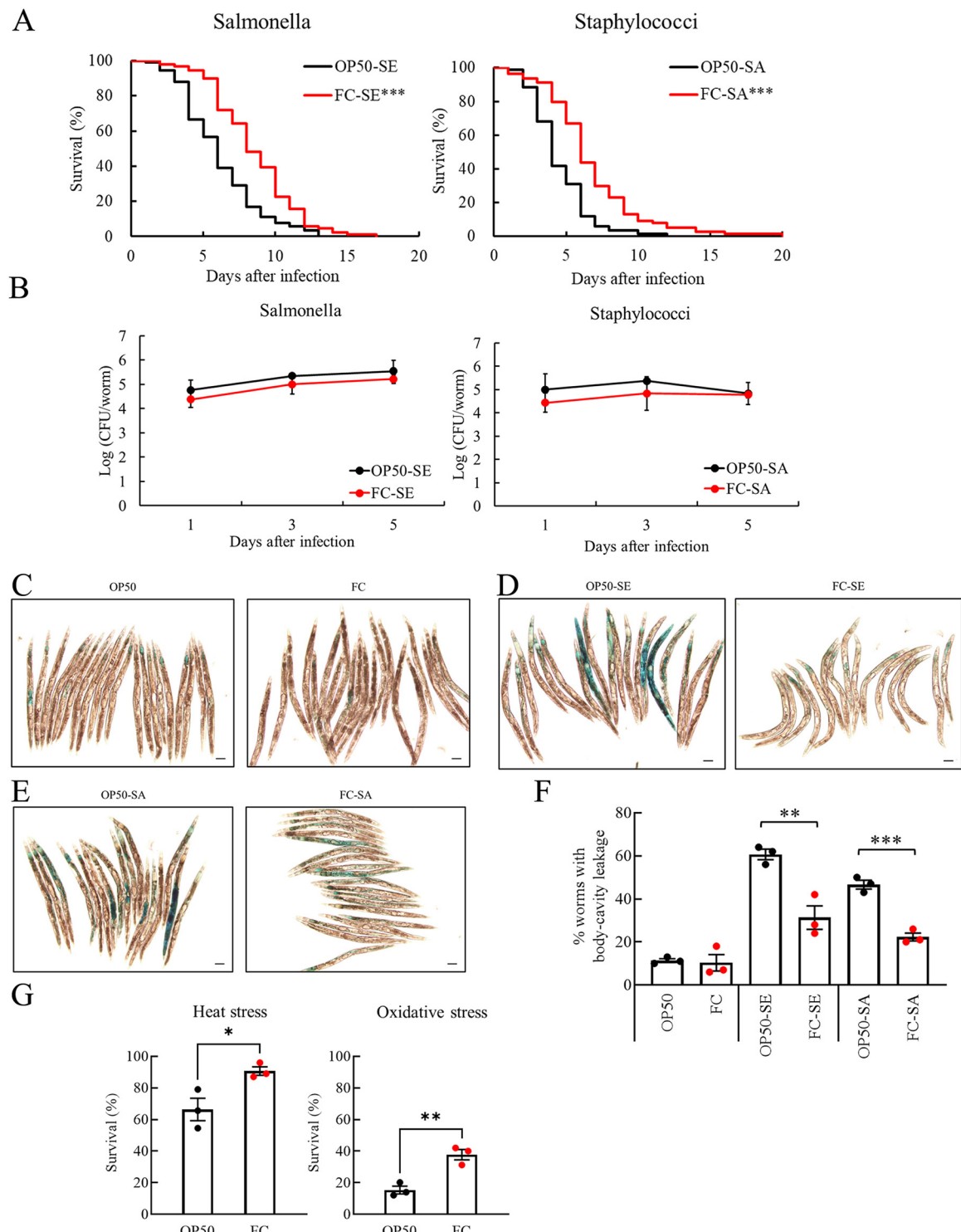

**FIG 3** Effects of FC on host defense of *C. elegans*. (A) Survival curves of *C. elegans* that were fed FC or OP50 for 4 days before infection with *Salmonella enterica* serovar Enteritidis (SE) or *Staphylococcus aureus* (SA). Asterisks indicate significant differences (***, $P < 0.001$) from control worms fed OP50 determined with log-rank test. OP50-SE worms, $n = 90$; FC-SE worms, $n = 89$; OP50-SA worms, $n = 84$; FC-SA worms, $n = 78$; trial no. 1 (Table S2) is shown as a representative graph. (B) The CFU of *S*. Enteritidis or *S. aureus* recovered from infected nematodes were obtained by plating whole-worm lysates 1, 3, and 5 days after the start of infection; $n = 5$ for each plot. Statistical analysis was performed by two-factor factorial ANOVA and Scheffe's *F* test. (C) The integrity of intestinal barrier function was assessed by examining the leakage of dye into the internal cavity (Smurf assay). Worms fed FC or OP50 for 9 days were examined. (D and E) After feeding young adult worms (0 days old) with FC or OP50, the 4-day-old nematodes were infected with *S*. Enteritidis (D) or *S. aureus* (E) and then stained with blue dye at 9 days old. Each scale bar indicates 100 μm. Magnification, ×40. (F) The percentage of worms showing body-cavity leakage was compared between FC-fed worms and those fed OP50 at day 9. *S*. Enteritidis or *S. aureus* infection caused more leakage than that in

A

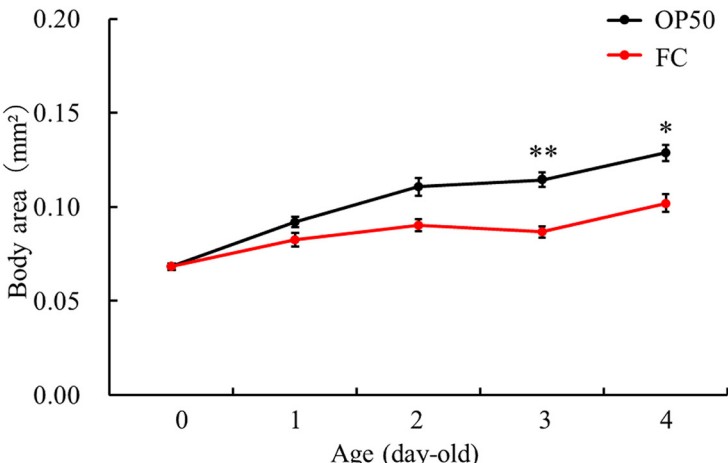

B

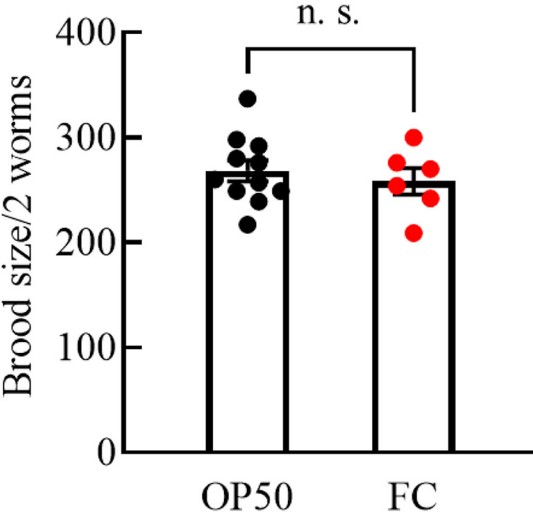

**FIG 4** Assessment of possibility of calorie restriction due to FC feeding. (A) Growth curves of worms fed OP50 or FC from day 0 to day 4. All results are shown as means ± standard error. Asterisks indicate significant differences (*, $P < 0.05$; **, $P < 0.01$) from control worms fed OP50 using two-factor factorial ANOVA and the Steel-Dwass test. $n = 10$ worms. (B) Comparison of brood size between worms fed FC and those fed OP50. OP50-fed worms, $n = 22$; FC-fed worms, $n = 12$. Error bars represent the standard error. Statistical analysis was performed with Student's $t$ test.

transcription factor, is the target of the insulin/insulin-like growth factor signaling pathway (IGF-1) and is activated to enhance host resistance when upstream IGF-1 signals decrease due to conditions such as dietary restriction. Another transcription factor, SKN-1, the target of the p38 MAPK (PMK-1) pathway, regulates the expression of xenobiotic detoxification genes (35). Indeed, mutant nematodes with defects in host defense-associated genes tended to have shorter life spans than the wild-type N2 (Fig. 5A). However, when mutant nematodes

**FIG 3** Legend (Continued)
noninfected worms; however, the leakage was less in worms fed FC before the infection. Error bars represent the standard error. Asterisks indicate statistically significant differences (**, $P < 0.01$; *** $P < 0.001$). Statistical analysis was performed with Student's $t$ test. (G) Influence of FC feeding on the susceptibility of worms to physical or chemical stresses. The young adult worms were fed FC or OP50 for 4 days and then exposed to heat or oxidative stress caused by juglone. Error bars represent the standard error. Asterisks indicate statistically significant differences (*, $P < 0.05$; **, $P < 0.01$) determined using Student's $t$ test and Welch's $t$ test.

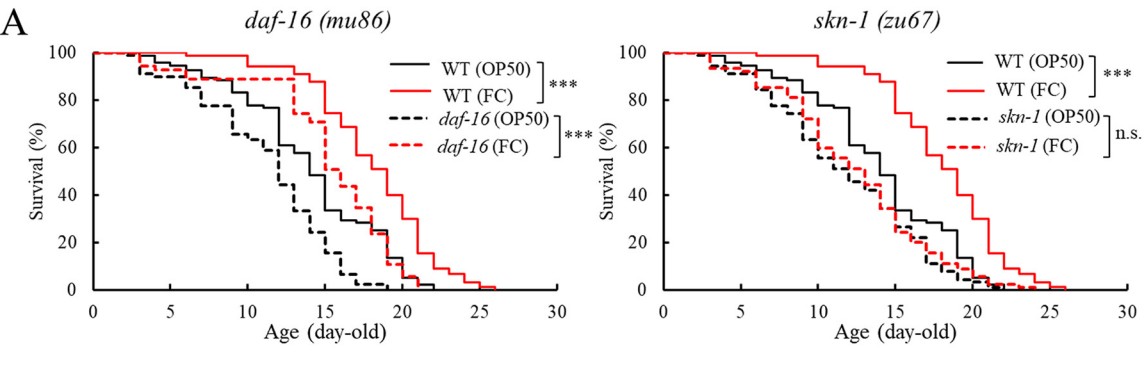

FIG 5 Identification of genes involved in antisenescence effects of FC. (A) Survival curves of *daf-16* (*mu86*), *skn-1* (*zu67*), and *pmk-1* (*km25*) mutant hermaphrodites fed with or without FC from 0 days old. The numbers of worms are indicated in Table S1. Asterisks indicate statistically significant differences (***, $P < 0.001$) compared to control worms fed OP50 using a log-rank test. (B) The CI of *daf-16* mutants fed OP50 or FC from day 0. The numbers of worms fed OP50 for 2, 3, and 4 days were 208, 211, and 220, respectively. The numbers of worms fed FC for 2, 3, and 4 days were 210, 218, and 217, respectively. Error bars represent the standard error. Asterisks indicate a statistically significant difference (*, $P < 0.05$) compared to control worms fed OP50 using two-factor factorial ANOVA and the Tukey-Kramer test. (C) The CI of *skn-1* mutants fed OP50 or FC for 4 days from the 0-day time point. The numbers of worms fed OP50 or FC were 218 and 221, respectively. Error bars represent the standard error. Statistical analysis was performed with Student's *t* test. (D) Survival curves of the *skn-1* (*zu67*) mutants infected by SE or SA. FC failed to enhance the host defense of *skn-1* mutants, unlike in wild-type worms. The numbers of worms are shown in Table S2. Statistical analysis was performed with a log-rank test.

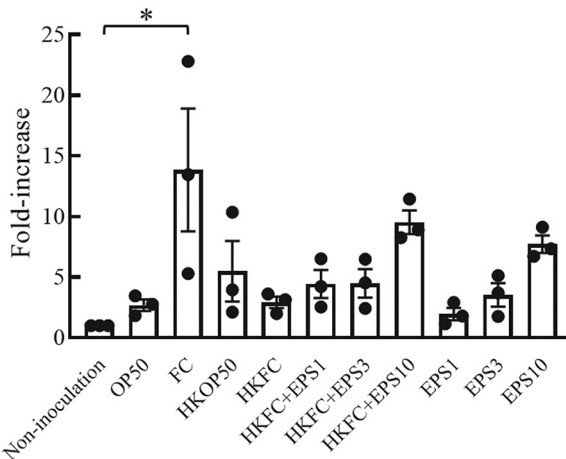

**FIG 6** Influence of FC on mammalian cells. HO-1 mRNA expression in J774.1 cells examined after incubation with viable FC or OP50, heat-killed FC (HKFC), or heat-killed OP50 (HKOP50) and with/without exopolysaccharide (EPS) for 24 h. EPS1, EPS3, and EPS10 denote that 1, 3, and 10 mg/mL EPS was inoculated into the cell culture media, respectively. Error bars represent the standard error. Asterisks indicate significant differences (*, $P < 0.05$) from noninoculated cells, using single-factor ANOVA and Scheffe's *F* test.

with defects in host defense-associated genes were fed FC, the bacteria extended the life span of *daf-16* and *pmk-1* mutants, but not that of *skn-1* mutants (Fig. 5A). Although the olfactory ability of *daf-16* mutants (Fig. 5B) was lower than that of wild-type N2 (Fig. 2B), 4-day-old FC-fed mutants showed a higher CI than the control (Fig. 5B). However, FC failed to improve the perception of *skn-1* mutants (Fig. 5C). Furthermore, the survival time of *skn-1* mutants after infection with *S.* Enteritidis or *S. aureus* was similar between OP50 and FC feeding (Fig. 5D): the *skn-1* mutants lost tolerance to both pathogens. It was suggested that FC exerted the beneficial effects via *skn-1*.

As SKN-1 is a homologue of nuclear erythroid 2-related factor 2 (Nrf2) in mammals, we investigated whether FC could enhance the Nrf2 antioxidant response pathway in mammalian cells. We measured the transcription of the HO-1 gene, which is regulated by Nrf2. The mRNA level of HO-1 increased by more than 14 times after inoculation with FC cells compared to that in noninoculated macrophages (Fig. 6). However, heat-killed bacteria could not produce this effect. The crude EPS induced the transcription of HO-1 in a dose-dependent manner. However, the effects of EPS were not statistically significant.

## DISCUSSION

In this study, we showed that *L. cremoris* strain FC, a Caspian Sea yogurt fermenter, extended the life span and health span of nematodes without affecting the brood size. FC could decrease AGE levels, delay senescence, and improve perception ability, making the nematodes resilient to infections and stressors. The increased autofluorescence from AGEs and the deterioration of olfactory ability were initially alleviated by probiotic feeding in either 6- or 4-day-old nematodes, respectively; unlike the conventional life span assay, these examinations rapidly determine whether probiotic bacteria and any chemicals exert longevity effects in *C. elegans*. Since *L. cremoris* has been accepted as a human probiotic, the present data could support the usage of *C. elegans* as a model organism for human probiotic studies.

A previous study suggested that the decline in CI could be attributed to sarcopenia, but not impaired cognition (36). However, in this study, comparatively young adult worms (2 to 4 days old) were used, with reportedly well-maintained integrity of their nervous system and muscles at this age (37). Indeed, in the chemotaxis assay, the number of worms moving out of the center from where they were settled first was similar between 2- and 4-day-old worms; thus, the locomotive function was not significantly impaired in 4-day-old worms. Furthermore, the response to odorants is low in worms aged more than 6 or 8 days due to functional deterioration of neuromuscular signaling (37). Thus, the decrease in the perception of the attractant might be responsible for the low CI in the worms fed OP50 until 4 days old. Even in mice,

audibility, another sensory neuron function, is lost earlier than other symptoms of senescence (27).

Even the international food standard OP50 can become pathogenic in old nematodes (9). Worms fed HKOP50 lived longer than those fed live OP50; the longevity afforded by FC was possibly attributed to its harmless nature. However, the nematodes fed FC lived longer than those fed HKOP50. Thus, the longevity by FC might not be simply attributed to its nonpathogenic property.

Donato et al. recently reported that extracellular polymeric substances produced *in vivo* by *Bacillus subtilis* were implicated in the prolonged longevity of nematodes (38). In this study, FC was recovered alive from the nematodes, and the nematodes fed live FC had better survival than those fed HKFC. Thus, we assumed that the EPS synthesized *in vivo* might contribute to the longevity effect of live FC. However, the EPS of FC failed to explain the longer life span of live FC-fed *C. elegans* than that of HKFC-fed worms because statistically significant extension by EPS was observed only once in three trials. Since biogenic amines, such as serotonin, could be involved in the antisenescence effects of probiotic bacteria (10, 39), live FC modifies the number of monoamines absorbed and influences senescence through the so-called gut-brain axis (40). FC might function as a psychobiotic bacterium in worms (41). Although it remains uninvestigated whether the organisms can produce bioactive amines, another probiotic, *Bifidobacterium dentium*, has been reported to stimulate serotonin synthesis in chromaffin cells (42). Employing such direct or indirect mechanisms, FC may alter neurotransmitter levels in worms.

Although the brood size of nematodes fed FC was normal, their body size and lipid amount were less than those of nematodes fed OP50. FC, potentially being a low-calorie food, is assumed to prolong the life span through the so-called caloric restriction. However, caloric restriction reportedly extends the life span of worms mainly via DAF-16 (43, 44), and Huang et al. showed that the longevity afforded by aspirin resulted in decreased fat storage via DAF-16 (45). In contrast, the effects of FC were independent of DAF-16 in this study. Further, it is difficult to extend the life span of worms by calorie restriction or maintaining their fertility; it is routine practice to sterilize worms with 5-fluorouracil or to use worms at the postreproductive adulthood stage because fertile worms escape from the plate upon calorie restriction (33, 34, 46). As a result of these contrasting findings, it is unlikely that the longevity of FC-fed worms can be explained only by the calorie restriction theory. Reduced fat storage might be due to antiobesity peptides or amino acids secreted by the bacteria rather than by calorie restriction (47, 48). It remains to be elucidated if FC could exert longevity effects without calorie restriction with such unknown factors.

Even under stresses such as exposure to heat or juglone and infections with *S. Enteritidis* or *S. aureus*, nematodes fed FC had better survival than those fed OP50. This resilience seems to be attributed to tolerance rather than resistance, because the number of pathogens recovered was similar between infected nematodes fed FC and those fed OP50. This is likely associated with the better epithelial barrier function in the worms fed FC, particularly via zonula occludens protein-1, which was indicated by less leakage in the intestinal barrier function assay (Smurf assay) (49); enhanced host resilience may also be due to biogenic monoamines, such as serotonin and dopamine, production of which is influenced directly or indirectly by FC (39).

For the longevity due to FC, SKN-1 was pivotal, whereas DAF-16, a well-known transcription factor that regulates genes promoting stress resistance and longevity (50), was not. SKN-1 belongs to the cap-n-collar family (CNC) of basic leucine zipper transcription factors and regulates the expression of xenobiotic detoxification genes (35). Reduction of the blue autofluorescence, an index for AGEs (28), in the worms fed FC might also be dependent on SKN-1 (51, 52). The mammalian transcription factor Nrf2 of the CNC family is homologous to SKN-1. In this study, FC caused higher HO-1 transcription in mouse macrophages than in nontreated cells. Darby et al. also reported the induction of Nrf2 by *L. cremoris* in a murine model (53). Beneficial effects, such as antioxidative and anti-inflammatory activities, have been well-reviewed and correlated

with human health by Yamamoto et al. (54). Thus, FC may promote health in both nematodes and mammals, as reported previously (15, 21–23, 26).

*L. cremoris* exhibited antisenescence effects in nematodes through supporting perceptive function and the suppression of AGEs by activating host defense via the transcription factor SKN-1. In addition, FC effectively activated Nrf2 in mammalian macrophages. Strains selected using the *C. elegans* model can therefore be candidates for human probiotics. The beneficial effects in worms cannot be sufficient to ensure probiotic function in humans; however, the positive results would be a prerequisite. In addition, this study suggests the potential use of *C. elegans* as a gut-brain axis model for screening psychobiotics.

## MATERIALS AND METHODS

**Nematodes.** *C. elegans* Bristol strain N2 and its loss-of-function mutant strains were kindly provided by the Caenorhabditis Genetics Center, University of Minnesota. All mutants—KU25 *pmk-1* (*km25*), EU1 *skn-1* (*zu67*), and CF1038 *daf-16* (*mu86*)—had been produced using ethyl methanesulfonate or UV irradiation. Nematodes were maintained and propagated on nematode growth medium (NGM) according to standard techniques (55). OP50 was used as the standard feed for nematode cultivation and was grown on tryptone soya agar (Nissui Pharmaceutical, Tokyo, Japan). Cultured bacteria (100 mg wet weight) were suspended in 0.5 mL M9 buffer, and 50 $\mu$L of the resulting bacterial suspension was then spread on peptone-free NGM (pfNGM) in 5.0-cm-diameter plates to feed the worms (56).

**Bacterial strains.** *Lactococcus cremoris* strain FC (a stock strain of Fujicco Co., Kobe, Japan) was used as a test food source for *C. elegans* and was aerobically cultured in M17 broth (Becton, Dickinson, Franklin Lakes, NJ, USA) or M17 agar (Becton, Dickinson) with 0.5% lactose at 25℃ for 48 h. *Salmonella enterica* subsp. *enterica* serovar Enteritidis strain SE1, originally isolated from a diarrheal specimen, was used as a Gram-negative pathogen. *Staphylococcus aureus* strain 96-55-17A was used as a Gram-positive pathogen. These pathogenic bacteria were reported as infection models using *C. elegans* (56) and were grown on tryptone soya agar at 37℃ for 24 h.

**Determination of *C. elegans* life span.** Eggs were recovered from adult *C. elegans* worms after exposure to sodium hypochlorite/sodium hydroxide solution, as previously described (57). The egg suspension was incubated overnight at 25℃ to allow hatching, and the resulting suspension of L1-stage worms was centrifuged at 156 × *g* for 1 min. After the supernatant was removed by aspiration, the remaining larvae were transferred onto fresh pfNGM plates covered with OP50 and incubated at 25℃. Pubescence was synchronized by allowing the worms to feed on OP50 for 2 days until maturation, as the reproductive system is known to regulate aging in *C. elegans* (58). Lifespan assays were performed by adding the young adult worms (nominal 0 days old) to each pfNGM plate covered with FC or OP50. The plates were incubated at 25℃, and the numbers of live and dead worms were scored every 24 h. At 25℃, the worms produce progeny that develop into adults within 3 days; therefore, identifying the original worms is difficult. Overestimation of the number of live worms was avoided by transferring the original worms daily to fresh pfNGM plates for 4 days until they completed their egg-laying phase at 4 days old. The worms were then transferred to fresh pfNGM plates every alternate day. A worm was considered dead when it failed to respond to a gentle touch with a worm picker. Worms that died from adhering to the wall of the plate were not included in the analysis. Life span assays are widely performed using NGM agar plates containing peptone, which facilitates the proliferation of the overlaid bacteria. However, it is reported that the composition of NGM influences the bacterial virulence and that the *in situ* production of metabolites by bacteria growing on the medium could also be fatal to nematodes (59). Thus, the possibility of bacterium-induced nematocidal effects from nutrients in the medium was avoided by performing the life span assays on pfNGM plates lacking peptone. Heat-killed bacteria were prepared at 100℃ for 10 min. Each assay was performed in duplicate and repeated more than twice, unless otherwise stated.

The mean life span (MLS) of adults was estimated using the following formula (60):

$$MLS = \frac{1}{N}\sum_{j}\frac{x_j+x_{j+1}}{2}d_j$$

where $d_j$ is the number of worms that died in the age interval ($x_j$, $x_{j+1}$), and $N$ is the total number of worms. The standard error (SE) of the estimated MLS were calculated using the following equation:

$$SE = \sqrt{\frac{1}{N(N-1)}\sum_{j}\left(\frac{x_j+x_{j+1}}{2}-MLS\right)^2 d_j}$$

The maximum life span of adults was calculated as the MLS of the top 15% longest-living worms in each group. Two technical replicates for each sample were used for the life span assay, and three biological replicates were analyzed for this study. Table S1 summarizes the data obtained from all experiments.

**Locomotory scoring of aging nematodes.** The motility of worms at different ages was examined using a scoring method described previously (57, 61). In brief, worms were classified as class A when they showed spontaneous movement or vigorous locomotion in response to prodding, class B when they did not move unless prodded or appeared to have uncoordinated movement, and class C when they moved only their head and/or tail in response to prodding. Dead worms were classified as class D. A minimum of 60 worms fed

OP50 or FC were scored. Two technical replicates for each sample were used for the locomotory assay, and three biological replicates were analyzed for this study.

**Measurement of autofluorescence as an aging marker in *C. elegans*.** We recently developed a method to measure autofluorescence as an aging marker for individual worms (28). In brief, 6-day-old worms fed OP50 or FC from 0 days were washed with M9 buffer and placed into 1.0 $\mu$L of M9 buffer on a 384-well black plate (Stem, Tokyo, Japan) covered with Saran Wrap (Asahi Kasei, Tokyo, Japan). The blank (M9 buffer only) data were checked three times for each well, considering the fluctuation across wells. Minimal detection limits and quantifiable limits were determined on the basis of blank data on each day as $\mu$ (mean of the blank) + 3.29$\sigma$ (standard deviation) and $\mu + \sqrt{2} \times 10\sigma$, respectively. The autofluorescence in the worm body was captured using a multimode grating microplate reader model SH-9000Lab (excitation, 340 nm; emission, 430 nm; Corona Electric, Ibaraki, Japan). After measurement, each worm was individually maintained on a 4-cm-diameter plate covered with OP50 or FC (2 mg/10 $\mu$L M9 buffer) at 25°C. The data on worms that died in 2 days were excluded because the autofluorescence could be due to death and not from AGEs (62). The assay was performed with more than 20 worms. Three biological replicates were analyzed for this study.

**Measurement of body size.** Adult worm size was determined as established previously (8). Young adult worms (0 days old) were placed on pfNGM plates covered with lawns of FC. The plates were incubated at 25°C, and the body size of live worms was measured every 24 h until they reached 4 days of age. Images of adult nematodes were captured using an STZ-161 microscope (Shimadzu, Kyoto, Japan) and a USB camera L-835 (Hozan, Osaka, Japan). Body size was analyzed using Adobe Photoshop Elements and ImageJ version 1.52a software developed by the National Institutes of Health. In this system, the area of a worm's projection was estimated automatically and used as the index of body size. Two biological replicates were analyzed for this study.

**Lipid staining in *C. elegans*.** Staining was performed as previously described with minor modifications (63). In brief, 4-day-old worms fed OP50 or FC for 4 days were washed three times with M9 buffer. The samples were fixed in 50% isopropanol in phosphate-buffered saline (PBS) for 15 min on ice. Oil red O stock solution (0.5 g/100 mL in isopropanol; Sigma-Aldrich, St. Louis, MO, USA) was diluted in distilled water (dH$_2$O) to 60% working solution and filtered using a 0.2-$\mu$m membrane. Fixed worms were incubated in working solution at 25°C for 20 min. Stained worms were washed with M9 buffer containing 0.5% Triton X-100 and mounted on glass slides using M9 buffer plus 0.5% Triton X-100 for imaging using an BX53 microscope equipped with a DP74 color camera (Olympus, Tokyo, Japan). The measured dye values were normalized to density values determined for the body size, as described above. Next, the dye values measured using ImageQuant TL version 8 (GE Healthcare, Chicago, IL, USA) and Adobe Photoshop Elements were normalized to density values per mm$^2$ of a worm's projection area. Two biological replicates were analyzed for this study.

**Preparation of EPS from FC.** EPS was isolated from fermented milk incubated at 26°C for 8 h with FC, as described previously (18). Trichloroacetic acid (TCA) was added at 4% (wt/vol) to the fermented milk to remove the precipitate and bacterial cells, and the resulting suspension was centrifuged at 18,890 $\times$ *g* for 20 min at 4°C. EPS was precipitated from the supernatant with 1.5 mL of chilled ethanol, collected using a spatula, dissolved in dH$_2$O, and precipitated at 18,890 $\times$ *g* for 20 min at 4°C with 4% TCA. The EPS was obtained by dialysis of the supernatant against dH$_2$O for 3 days and subsequent lyophilization.

EPS was administered in the form of a liposome, so that *C. elegans* could ingest hydrophilic substances (30). The liposome was prepared and administered as previously described, with minor modifications (30). In brief, the EPS was dissolved in dH$_2$O (15 mg/mL). Next, L-$\alpha$-phosphatidylcholine was added to the solution at 40 mg/mL, and liposomes were produced by mixing the solution at 65°C and passing it through a Nuclepore track-etched membrane (pore size, 5.0 $\mu$m; Whatman, Newton, MA, USA) using a miniextruder (Avanti Polar Lipids, Alabaster, AL, USA). For comparison, liposomes containing dH$_2$O were used as a control. Nematodes were fed 25 $\mu$L OP50 (400 mg/mL) and 25 $\mu$L liposomes containing EPS on pfNGM from 3 days of age. Two technical replicates for each sample were used for the life span assay, and three biological replicates were analyzed for this study. Table S1 summarizes the data obtained from all experiments.

**Chemotaxis assays.** This assay was performed as described previously (64), with modifications. Synchronized wild-type worms and *skn-1* and *daf-16* mutants were fed OP50 or FC from 0 days old. The 2- to 4-day-old worms were collected and assayed on 90-mm pfNGM plates. Next, 1 $\mu$L of 1.0 M sodium azide along with 1 $\mu$L of the odorant 0.1% benzaldehyde in 100% ethanol was dropped onto the center of attractant regions A and D (2 cm in diameter) on the plates. At the center of regions B and C, located at the opposite end of the attractant regions, 1 $\mu$L of 100% ethanol as the control and 1 $\mu$L of sodium azide were dropped. Next, about 50 to 75 worms washed with M9 buffer were immediately transferred to the center of the plate; only worms that moved spontaneously and showed vigorous locomotion in response to prodding were used. The assay plates were incubated at 25°C for 2 h, and then the number of worms in each quadrant was scored. The CI was calculated using the formula CI = (number of worms in both attractant quadrants – number of worms in both control quadrants)/total number of worms on the assay plate. Worms that adhered to the plate wall were not included in the analysis. Three technical replicates for each sample were used for the chemotaxis assay, and three biological replicates were analyzed for this study.

**Influence of feed on resistance against pathogenic bacterial infection.** Resistance against pathogenic bacterial infections was determined as previously described (8). The young adult worms (0 days old) were assigned to either a control group that was fed OP50 or to a group that was fed FC for 4 days. The 4-day-old worms were then transferred onto *S.* Enteritidis or *S. aureus* lawns. Each group was incubated at 25°C. The numbers of live and dead worms were scored every 24 h. Survival rates were compared between the groups of worms grown on different feeds before the *S.* Enteritidis or *S. aureus* infection. Two technical replicates for each sample were used for the survival assay, and three biological replicates were analyzed for this study. Table S2 summarizes the data obtained from all experiments.

**Measurement of the number of bacteria in nematodes.** The number of bacterial cells in the nematodes was determined according to a previously described method (65) with modifications. The surface bacteria were killed by immersing more than five worms in 500 $\mu$L gentamicin solution (1 mg/mL) for 30 min in a 0.5-mL microtube. After the worms were washed five times in M9 buffer, each nematode was placed in a microtube containing 50 $\mu$L M9 buffer and mechanically disrupted using a mini cordless grinder (Funakoshi, Tokyo, Japan). The volume was adjusted to 1 mL using M9 buffer. Before culturing on agar, the homogenate solution was diluted in M9 buffer, which was plated on tryptone soya agar for culturing *S*. Enteritidis and *S. aureus* at 37°C. All plates were incubated for 48 h. Two biological replicates were analyzed for this study.

**Smurf assay for intestinal barrier function.** The integrity of the intestinal barrier function was examined using the method described by Kim and Moon (49). Worms were infected by following the infection method as described previously. Infected or noninfected worms (9 days old) on pfNGM agar were transferred to S-basal medium containing heat-killed OP50 mixed with acid blue 9 as a blue food dye (5% [wt/vol] in S-basal solution with cholesterol; Tokyo Chemical Industry Co., Tokyo, Japan) at 25°C for 3 h. Worms were washed in M9 buffer five times before they were anesthetized in M9 buffer containing 500 mM sodium azide. Subsequently, the worms were observed on the agar pad for the presence of blue dye in the body cavity using an Olympus BX61 microscope and USB camera WRAYCAM-VEX120; (Wraymer, Inc., Osaka, Japan). Thereafter, the stained worms were counted to determine body cavity leakage. The assay was performed with a minimum of 30 worms, and 3 biological replicates were analyzed for this study.

**Stress resistance assays.** In this method, 4-day-old worms fed OP50 or FC from the 0-day-old stage were exposed to heat and various oxidative stresses. Thermal tolerance was assessed by incubating worms at 35°C for 7 h, and the survival rate was calculated after the worms were recovered on pfNGM seeded with OP50 or FC at 25°C for 24 h. An oxidative stress assay was conducted by transferring 4-day-old worms onto pfNGM containing 250 $\mu$M juglone (a reactive oxygen species generator) for 2 h, and viability was scored after a 15 h recovery period on normal pfNGM seeded with OP50 or FC. Juglone was dissolved in 100% ethanol, and the assays were performed according to the method described previously (66, 67) with some modification. For other oxidative stress assays, worms were placed onto pfNGM containing 1.0 mM paraquat or in 24-well tissue culture plates containing 7.0 or 9.0 mM cupric chloride added to K-medium (53 mM NaCl, 32 mM KCl) (68), and the numbers of live and dead worms were scored either daily or every hour, respectively. The survival of worms was determined using touch-provoked movements. Worms were scored as dead when they failed to respond to mechanical stimuli with a worm picker. The assay was performed on 100 or more worms. Two technical replicates for each sample were used for the heat or stress assay, and three biological replicates were analyzed for this study.

**Brood size.** Brood size was determined as previously established (57). Eggs isolated with a sodium hypochlorite or sodium hydroxide solution were allowed to develop into young adults on pfNGM plates coated with OP50 at 25°C. Two hermaphrodites were selected and transferred to a pfNGM plate covered with a lawn of OP50 or FC. The parental animals were transferred every 24 h to fresh pfNGM plates until the end of the reproductive period. The resulting progeny were left to develop for 3 days, and the progeny number was then determined. Biological replicates ($n = 3$), with over 6 to 11 technical replicates each of 2 animals were used for data analysis.

**Cell culture and mRNA expression.** Initially, the Hep G2 cell line (BPS Bioscience, San Diego, CA, USA) containing a luciferase gene under the control of an antioxidant response element was used. Although FC was assumed to activate the antioxidant response element of this reporter cell line, it failed unexpectedly, potentially due to poor bacterial ingestion by hepatic cells. Thus, the murine macrophage cell line J774.1 (JCRB0018; JCRB Cell Bank, Osaka, Japan) was used instead. These cells were grown in RPMI 1640 (Wako, Osaka, Japan), supplemented with 0.5% (vol/vol) StemSure 10-mmol/l 2-mercaptoethanol solution (Wako) and 10% (vol/vol) fetal bovine serum (no. F7524; Sigma-Aldrich), as the cell medium. Cells were grown to approximately 80% confluence in 25-cm$^2$ polystyrene tissue culture flasks at 37°C in a 5% $CO_2$ incubator. Before bacterial treatment, the cells were transferred to a 24-well plate ($5 \times 10^5$ cells/500 $\mu$L cell medium/well; Thermo Fisher Scientific, Waltham, MA, USA) overnight. The bacteria were collected in tubes and washed with PBS. For heat-killed bacteria, bacteria were boiled at 100°C for 10 min and then adjusted to 100 mg/mL with cell medium. The J774.1 cells were cultured with live or heat-killed bacteria of either OP50 or FC (0.5 mg/500 $\mu$L cell medium) for 24 h. For EPS, cells were treated alone (EPS: 1, 3, or 10 mg/mL) or mixed with HKFC (0.5 mg/500 $\mu$L cell medium). Total mRNA was extracted from either the bacterium- and/or EPS-treated cells or from untreated cells using the NucleoSpin RNA kit (Macherey-Nagel, Duren, Germany) according to the manufacturer's instructions. cDNA was synthesized using ReverTra Ace quantitative PCR real-time (qPCR RT) master mix with genomic DNA (gDNA) remover (Toyobo, Osaka, Japan). The transcription levels of the HO-1 gene were detected using the Luna Universal qPCR master mix (New England Biolabs Japan, Tokyo, Japan). Real-time PCR was performed using Mastercycler ep realplex$^2$ (Eppendorf, Hamburg, Germany). Information on the primers used is shown in Table 1. Target mRNA expression was normalized to the expression of a reference mRNA (actb, encoding the housekeeping protein $\beta$-actin), and the fold change was calculated on the basis of

**TABLE 1** Primers for qPCR

| Gene | Primers |
| --- | --- |
| $\beta$-actin | (Forward) 5′-CCCATCTATGAGGGTTACGC-3′ |
| | (Reverse) 5′-TTTAATGTCACGCACGATTTC-3′ |
| HO-1 | (Forward) 5′-CACGCATATACCCGCTACCT-3′ |
| | (Reverse) 5′-CCAGAGTGTTCATTCGAGCA-3′ |

the threshold cycle (ΔΔ*CT*) method (69). Three technical replicates for each sample were used for the qPCR assay, and three biological replicates were analyzed for this study.

**Statistical analysis.** Nematode survival was calculated using the Kaplan-Meier method, and survival differences were tested for significance using the log-rank test. Autofluorescence values were analyzed using the Mann-Whitney *U* test. Oil red O stain values, intestinal integrity with Smurf assay, brood size, heat stress, oxidative stress induced by juglone, and CI values for the *skn-1* mutant were compared using Student's *t* test and Welch's *t* test. Body size and CI values for wild-type N2 and *daf-16* mutants and the number of bacterial cells in *C. elegans* were analyzed using a two-factor factorial analysis of variance (ANOVA) and Steel-Dwass test, and the Tukey-Kramer test or Scheffe's *F* test was used for multiple comparisons. The mRNA expression of the cells was analyzed using single-factor ANOVA and Scheffe's *F* test. Where significance was observed, data were classified as follows: *, $P < 0.05$; **, $P < 0.01$; and ***, $P < 0.001$. All statistical analyses were performed using Microsoft Excel supplemented with the add-in software plus Statcel 3 and 4 (OMS, Tokyo, Japan).

**Data availability.** The data sets generated during the current study are available from the corresponding author on reasonable request.

## SUPPLEMENTAL MATERIAL

Supplemental material is available online only.
**SUPPLEMENTAL FILE 1**, PDF file, 0.1 MB.

## ACKNOWLEDGMENTS

The nematodes used in this study were kindly provided by the Caenorhabditis Genetics Center, which is funded by the NIH Office of Research Infrastructure Programs (P40 OD010440).

This study was supported in part by a Grant-in-Aid for Young Scientists (no. 19K15788) from the Japan Society for the Promotion of Science (T.K.) and the Institute for Fermentation (no. G-2020-3-057; Osaka, Japan) (T.K.).

We thank Editage (www.editage.com) for English language editing.
We declare no competing interests.

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
