## [Reviewer comments · Microbiology Spectrum]

Microbiology Spectrum

Prolonged Lifespan, Improved Perception, and Enhanced Host Defense of *Caenorhabditis elegans* by *Lactococcus cremoris* subsp. *cremoris*

Tomomi Komura, Asami Takemoto, Hideki Kosaka, Toshio Suzuki, and Yoshikazu Nishikawa

Corresponding Author(s): Yoshikazu Nishikawa, Tezukayamagakuin University

Review Timeline:

Submission Date:	September 22, 2021
Editorial Decision:	November 2, 2021
Revision Received:	February 24, 2022
Editorial Decision:	March 10, 2022
Revision Received:	March 31, 2022
Accepted:	April 19, 2022

Editor: Cheng-Yuan Kao

Reviewer(s): The reviewers have opted to remain anonymous.

Transaction Report:

DOI: <https://doi.org/10.1128/spectrum.00454-21>

November 2, 2021

Prof. Yoshikazu Nishikawa
Tezukayamagakuin University
Faculty of Human Sciences
4-2-2 Harumidai
Sakai 5900113
Japan

Re: Spectrum00454-21 (Prolonged Lifespan, Improved Cognition, and Enhanced Host Defense of *Caenorhabditis elegans* by *Lactococcus cremoris* sp. nov.)

Dear Prof. Yoshikazu Nishikawa:

Thank you for submitting your manuscript to Microbiology Spectrum. When submitting the revised version of your paper, please provide (1) point-by-point responses to the issues raised by the reviewers as file type "Response to Reviewers," not in your cover letter, and (2) a PDF file that indicates the changes from the original submission (by highlighting or underlining the changes) as file type "Marked Up Manuscript - For Review Only". Please use this link to submit your revised manuscript - we strongly recommend that you submit your paper within the next 60 days or reach out to me. Detailed information on submitting your revised paper are below.

Link Not Available

Sincerely,

Cheng-Yuan Kao

Journals Department
Reviewer comments:

Reviewer #1 (Comments for the Author):

Komura et al. have presented their study on the evaluation of *Lactococcus cremoris* sp. nov. strain FC (FC) as a contributor to the improvement of host defence, chemotactic activity and extended lifespan. The study was performed on *C. elegans*, a well described model of aging and senescence. A number of assays were performed on both wild type N2 and selected mutant worms fed with either *E. coli* OP50 (as control) or the FC strain. The authors state that live FC successfully extended the health span, enhanced host defence towards *Salmonella enterica* and *Staph aureus*, and maintained the cognitive ability of the nematodes. To further dissect the involvement of signalling pathways in the beneficial effects of FC on the worms, the authors examined the effect of FC on *daf-16*, *pmk-1* and *skn-1* mutant worms. Their findings suggest that FC affects longevity through the SKN-1 pathway. In addition, the authors also propose a role for extrapolymeric substances (EPS) produced by FC in vivo towards the beneficial features of FC.

I commend the authors on the amount of data generated from this study; however, I also have some concerns that need to be addressed to improve the manuscript. These concerns and additional suggestions are noted below:

1. The abstract begins with a description of what the study was about without a preamble to describe why the FC strain was selected for this evaluation when many other probiotic (LAB) have already been evaluated, including work from the authors. There has to be 1-2 sentences on the background and motivation for the study. Similarly, the abstract lacks a concluding sentence to put the findings in the context of new information or additional benefits over previous studies.
2. I feel that the description of the results can be improved. Every sample or parameter that is presented in a figure or a table is there because the data contributes to the overall findings. Nonetheless, the authors merely state what they feel is important and a lot of information is not presented or elaborated upon. Furthermore, as the methodology is only described at the end of the paper and the accompanying figure legends are extremely long and overly descriptive, the reader has to keep referring to the relevant methodology and figure legend to know what are the different samples or even a brief explanation of what was done. E.g in Fig 1 HKOP50 and HKFC are not mentioned in the text.
3. Table 2: Datasets for individual replicates are not normally presented in the main text and should be submitted as Supplementary Information. A representative median and maximum life span for each treatment should be presented. Similar comment for Table 3.
4. Fig 1G and the conclusions in the text - I disagree that EPS prolonged the worm lifespan as only a single experiment out of 3 replicate experiments showed any extended lifespan. Furthermore, I don't see any evidence that FC does indeed produce EPS or that the method used to produce the EPS from milk did indeed produce EPS without any significant contamination of other milk proteins that may also have been precipitated when using TCA. Unless, there is proof available, any reference to the contribution of EPS in extending lifespan does not have a strong justification to be retained in this submission or needs to significantly toned down.
5. I found the time points a little confusing when the data is described or discussed in the context of the age of the worm but the figures refer to days of treatment which don't exactly correspond - one needs to constantly remember that day 0 treatment correlates to 3-day old nematodes.
6. As noted in #4 above, lines 126-129 require further details (brief description of the experiment) and data interpretation. Similarly Fig. 3C, 3D, 3E and 3F. Line 143-145, where is the data for CuCl and paraquat?
7. A reader may potentially not understand the proposed link between FC and calorie restriction especially when it is stated that "FC is a low calorie food" with no reference to support this statement and again, how is this related to brood size? Related to this, can the authors explain why they did not choose to treat the worms to prevent or reduce egg laying so as not to interfere with the estimation of lifespan or even survival.
8. Data related to Fig. 5: Fig 5B & 5C - Why are there different treatment time data for CI of daf-16 but not skn-1? Fig. 5D: The data should also include N2 + SE or SA
9. The authors begin the discussion by stating on lines 183-185 - "Interestingly", FC made the nematodes resilient to infections and stressors, decreased AGEs levels, delayed senescence, and improved cognitive ability. I am not sure why this is "interesting" as this fact for various LAB or probiotics is well established.

Reviewer #2 (Comments for the Author):

Review 251886 Spectrum Komura et al

Komura et al. examined the effects of lactococci (LAB) on aging, lifespan and sensorimotor functions of *C. elegans*. They concluded that lactococci extend *C. elegans* life span likely through the Nrf2 transcription factor SKN-1. Cultivation of *C. elegans* on LAB slows down the accumulation of autofluorescent materials in the worm intestinal tissues, as well as delays the decline in chemotaxis functions. They provided some evidence that the exopolysaccharides (EPS) of LAB could play a role in some of the anti-aging effects of this probiotic microbe. The authors have invested significant efforts in a number of assays to show the beneficial effects of LAB on *C. elegans*, which should be acknowledged. However, the study is also complicated by a number of conceptual and technical concerns, as detailed below.

Major comments:

1. The authors argue that it is unlikely that caloric restriction contributes to the pro-longevity effects of FC. However, several lines of evidence suggest that this possibility is real, including (1) FC-fed worms are smaller, and (2) FC-fed worms accumulate significantly less fat. The authors are recommended to study the accumulation of yolk protein in the treated hermaphrodites as a way to further evaluate the nutritional status of FC-fed worms. Citing intact brood size as a lack of caloric restriction is not appropriate, as this does not rule out caloric reduction (and the authors did not provide data of brood size, although described in the Methods section how this was performed). Or at least they should rephrase this part and leave the possibility open for further in-depth investigation.

2. I do not support the use of cognitive function in this paper, as only chemotaxis to benzaldehyde was examined. *C. elegans* displays innate preference to benzaldehyde, and this is regarded as a sensory behavior but not cognition - cognition usually refers to more sophisticated behaviors such as learning and memory. To extrapolate the results of benzaldehyde chemotaxis to other behavioral aspects is not warranted unless more behavioral assays are conducted.

3. The effects of EPS on longevity are questionable, since only one in three experiments showed slight but significant extension of life span compared to that of the control. The last sentence of the abstract ("both FC and EPS can affect longevity...") should be revised. I need to further point out that the effects of EPS on life span are not examined in the *skn-1* mutant. Therefore, this statement in the abstract is unsupported and misleading.

4. Page 9, Line 174-177: Since EPS at a concentration of 10 mg/ml increases the expression of HO-1 to a level similar to that by heat-killed FC + 10 mg/ml EPS, it suggests that EPS at higher concentration could increase the expression of HO-1 without FC. Therefore, the conclusion that "it worked synergistically with heat-killed FC cells" is not supported by the data and must be revised. Or additional statistics should be provided to justify the authors' claim.

5. Fig. 5A: The life span of the *daf-16* and *skn-1* mutants looks similar to that of the wild type, which is different from what most *C. elegans* studies had found. This is very strange. The authors should include N2 for experiments in Fig. 5A.

6. *skn-1(zu67)* is zygotic lethal, and one cannot have viable *zu67* homozygous adult worms for life span assays. The authors need to explain how life span experiments with *skn-1(zu67)* were performed.

Minor points:

1. P5, Line 79: "gram positive" should be "Gram-positive". Please correct this wherever it applies.

2. P6, Line 113: What does "ovary" in *C. elegans* mean?? This is confusing. There is something called uterus but not ovary in *C. elegans*.

3. P7, Line 129: "7 days of age" refers to worm age dated from newly hatched L1 larvae (= Day 4 adult?)? But the standard in *C. elegans* aging research community refers to days in the adulthood as the age of the worm. This should be clearly indicated.

4. P7, Line 144-145: indicate "data not shown" or provide the data for cupric chloride or paraquat. Why are FC-fed worms more resistant to juglone but not to paraquat, since both induce oxidative stress?

5. Page 9, Line 170: change "similar to" to "a homologue of the"

6. Page 11, Line 218: change "S. Enteritidis" to "enteritidis" (lower case italicized)

7. Bar graphs should reveal all individual data points (Figs 1F, 2B, 3F, 3G, 4B, 5B, 5C, 6)

8. Fig. 1A, 1B, 1G, 3A, 5A, 5D: the life span curves look weird with those thick vertical lines. In Fig. 1B, the vertical lines of the life span curves have multiple circles. Please correct these.

9. Fig. 1C: Statistics? Incorporate A/B/C classes into the labels of locomotion groups.

10. Fig. 1D: please show the error ranges (box plots with quartiles will be appropriate) in the data and also representative fluorescent images.

11. Fig. 1F: the unit at the Y-axis should be simplified.

12. Fig. 2: Please indicate benzaldehyde as the attractant in Figure 2 and the legend.

13. Fig. 5: please use the standard *C. elegans* nomenclature of mutant strains, including the allele name. Indicate in the B panel that this is the *daf-16* mutant, and in the C and D panels that it is the *skn-1* mutant. Also indicate the age of tested animals in Fig. 5C, not just in the legend.

14. Fig. 6: add unit (mg/ml) to the EPS label at the X-axis.

15. Remove redundancy of descriptions in the Figure legends. Some legends contain technical details that should be moved to the Methods section.

Staff Comments:

Preparing Revision Guidelines

Please return the manuscript within 60 days; if you cannot complete the modification within this time period, please contact me. If you do not wish to modify the manuscript and prefer to submit it to another journal, please notify me of your decision immediately so that the manuscript may be formally withdrawn from consideration by Microbiology Spectrum.

This submission by Komura et al on Prolonged Lifespan, Ameliorated Cognition, and Improved Host Defense of *Caenorhabditis elegans* by *Lactococcus lactis* subsp. *cremoris* set out to evaluate if the lactic acid bacteria *Lactococcus lactis* subsp. *cremoris* strain FC (FC) could improve host defenses and cognitive ability and extend lifespan using the *Caenorhabditis elegans* model of aging and infection.

The authors performed a number of different assays which showed that live FC were able to extend the health span, enhance host defense, and improve cognitive ability of the nematodes. Furthermore, senescence of sensory neurons was more stable in worms fed FC and significantly higher than that of the control worms. Worms fed FC were also able to resist the killing effects of *Salmonella enterica* serovar Enteritidis and *Staphylococcus aureus* infection for a longer period of time compared to worms fed on OP50. Using different *C. elegans* mutants, the authors proposed a key role of the transcription factor SKN-1 which regulates the transcription of heme oxygenase-1 whereby HO-1 mRNA expression in murine macrophages was increased >5 times after treatment with FC. The study concludes that FC affects longevity *via* the SKN-1/Nrf2 pathway in both nematodes and mammalian cells.

This is an interesting study and I commend the authors on the amount of data generated. Nonetheless, I feel that the significance of the data has not been elaborated on sufficiently within the results and discussion sections. The authors state "In this study, we evaluated whether strain FC could ameliorate host defenses, retard the accumulation of advanced glycation end-products (AGEs) and deterioration of cognitive ability, and prolong the lifespan of *C. elegans*." The authors in their earlier study (Ikeda et al. 2007 Appl. Environ. Microbiol. **73**:6404-6409) already demonstrated some aspects of this investigation with different LAB species. It is not clear what is the advantage of extending the study on the FC strain.

Other comments:

1. I suggest that the term OP50 is used instead of OP, which is the general practice of researchers working on *C. elegans*.
2. Lines 55-62 – these facts require at least one reference to be quoted
3. Lines 104-105: how do you know that the heat killed OP50 were less virulent – provide an appropriate reference that attests to this point.
4. Most of the experiments using anaerobic FC cells were performed only once – how were the authors able to conduct statistical analysis with only data from one experiment?
5. Figure 2: I believe the data for OP50 (live) was not obtained in that particular experiment but rather taken from earlier experiments and incorporated into Figure 2. If this is indeed so, then it has to be removed and no comparison can be made to that data.
6. The study used 10mg/ml of bacteria - why? Most studies use a fixed volume or number of cells. As lactobacilli are larger (and presumably heavier) than OP50, would this have any bearing on the observations?
7. Figure 1D: what is the median value of both data sets – how do you determine the significant difference without providing the median?
8. Figure 3B: again there is data from only one experiment but statistical analysis has been performed?
9. What was the significance of the experiments utilising the EPS preparation? Lines 121-122: my observation based on the data in Table 2 is that there is no significant difference in EPS/liposome compared to H2O2/liposome.
10. The survival on *pmk-1* mutants (Table 2) is not significant for FC vs OP50 for 2 of the replicate experiments.

11. I think the data presented in Figure 5A should include the wild type N2 worms as well.
12. Lines 231-254 within the Discussion: I don't see the significance / relevance to the context of this study.
13. The magnification of the DIC pictures should be included in the figure legends.

Reviewer comments:

Reviewer #1 (Comments for the Author):

Komura et al. have presented their study on the evaluation of *Lactococcus cremoris* sp. nov. strain FC (FC) as a contributor to the improvement of host defence, chemotactic activity and extended lifespan. The study was performed on *C. elegans*, a well described model of aging and senescence. A number of assays were performed on both wild type N2 and selected mutant worms fed with either *E. coli* OP50 (as control) or the FC strain. The authors state that live FC successfully extended the health span, enhanced host defence towards *Salmonella enterica* and *Staph aureus*, and maintained the cognitive ability of the nematodes. To further dissect the involvement of signalling pathways in the beneficial effects of FC on the worms, the authors examined the effect of FC on *daf-16*, *pmk-1* and *skn-1* mutant worms. Their findings suggest that FC affects longevity through the SKN-1 pathway. In addition, the authors also propose a role for extrapolymeric substances (EPS) produced by FC in vivo towards the beneficial features of FC.

I commend the authors on the amount of data generated from this study; however, I also have some concerns that need to be addressed to improve the manuscript. These concerns and additional suggestions are noted below:

First of all, we would like to thank you for reviewing our manuscript. Considering the comments from both the reviewers, we have revised most of the figures based on the age of adulthood to reduce the confusion that arose in the previous version of the manuscript where both the day of treatment and the age were used. For clarity, we have now defined the age of young adult as nominal 0-day-old when worms reached the third day after hatching. All experiments were performed on 0-day-old adults with FC feeding. Consequently, all description of ages were revised throughout the manuscript.

1. The abstract begins with a description of what the study was about without a preamble to describe why the FC strain was selected for this evaluation when many other probiotic (LAB) have already been evaluated, including work from the authors. There has to be 1-2 sentences on the background and motivation for the study. Similarly, the abstract lacks a concluding sentence to put the findings in the context of new information or additional benefits over previous studies.

The reason for using the FC strain *L. cremoris* in this study is as follows. A variety of lactic acid bacteria have been used as commercial starters of yogurt worldwide: *Bifidobacterium animalis*, *B. bifidum*, *B. infantis*, *B. longum*, *Lactobacillus acidophilus*, *L. bulgaricus*, *L. delbrueckii* subsp. *bulgaricus*, *L. gasseri*, *L. paracasei*, *L. rhamnosus*, and *Streptococcus thermophilus*. *Caenorhabditis elegans* has been recognized as a convenient organism to examine the beneficial functions of probiotics since our first report in 2007. However, bacteria which act as probiotics in nematodes may not necessarily have probiotic function in humans. Caspian-sea yogurt is one of the most popular yogurts in Japan, and its starter culture, *Lactococcus cremoris*, has been recognized for its probiotic function in mammals including humans. If *L. cremoris* could act as a probiotic in nematodes, it reinforces the possibility that the beneficial data in *C. elegans* model could be extrapolated to mammals, including humans. Considering your comment, we have revised the abstract and concluded that FC, a human probiotic bacteria, was also beneficial in *C. elegans* for retardation of

senescence.

2. I feel that the description of the results can be improved. Every sample or parameter that is presented in a figure or a table is there because the data contributes to the overall findings. Nonetheless, the authors merely state what they feel is important and a lot of information is not presented or elaborated upon. Furthermore, as the methodology is only described at the end of the paper and the accompanying figure legends are extremely long and overly descriptive, the reader has to keep referring to the relevant methodology and figure legend to know what are the different samples or even a brief explanation of what was done. E.g in Fig 1 HKOP50 and HKFC are not mentioned in the text.

Lines 108-114. We have duly explained the day of feeding and ages, and what HKFC and HKOP50 mean.

We have also revised the text in the Materials and Methods section to describe the details rather than in Figure Legends.

3. Table 2: Datasets for individual replicates are not normally presented in the main text and should be submitted as Supplementary Information. A representative median and maximum life span for each treatment should be presented. Similar comment for Table 3.

Lines 133, 328, 397, 423, 734, 740, 757, 774, 802, and 813 indicate the suggested change. Tables 2 and 3 were moved to constitute Supplementary Tables 1 and 2, respectively.

4. Fig 1G and the conclusions in the text - I disagree that EPS prolonged the worm lifespan as only a single experiment out of 3 replicate experiments showed any extended lifespan. Furthermore, I don't see any evidence that FC does indeed produce EPS or that the method used to produce the EPS from milk did indeed produce EPS without any significant contamination of other milk proteins that may also have been precipitated when using TCA. Unless, there is proof available, any reference to the contribution of EPS in extending lifespan does not have a strong justification to be retained in this submission or needs to significantly toned down.

Lines 55, 132–134, 191–192, and 225–227 are worth mentioning here. Considering your comment, we have deleted the statements suggesting that EPS may have beneficial effects. Further, we have made reference to EPS as crude EPS.

5. I found the time points a little confusing when the data is described or discussed in the context of the age of the worm but the figures refer to days of treatment which don't exactly correspond - one needs to constantly remember that day 0 treatment correlates to 3-day old nematodes.

Line 109–110. To make ourselves clear, we wrote that day 1 of FC feeding meant the third day (nominal 0-day-old) after hatching: 0-day-old worms were used for the FC feeding. Therefore, “day 1 of treatment” means the first day of adulthood. However, we have revised most of the data based on the age of adulthood to avoid the confusion that arose in the previous version of the manuscript.

6. As noted in #4 above, lines 126-129 require further details (brief description of the experiment) and data interpretation. Similarly Fig. 3C, 3D, 3E and 3F. Line 143-145, where is the data for CuCl₂ and paraquat?

Lines 132–134. Considering your comment, we have deleted the description that EPS may have beneficial effects.

Lines 135–140, 147–154. According to your suggestion, we tried to explain the results clearly.

Lines 157–159. FC failed to protect the worms from CuCl₂ and paraquat, and we did not show the data. If negative data are required to be presented, we could present the data in supplementary figures. Alternatively, we could delete the sentence if the reviewer would agree.

7. A reader may potentially not understand the proposed link between FC and calorie restriction especially when it is stated that "FC is a low calorie food" with no reference to support this statement and again, how is this related to brood size? Related to this, can the authors explain why they did not choose to treat the worms to prevent of reduce egg laying so as not to interfere with the estimation of lifespan or even survival.

Lines 162–169, and 234–247. Dietary restriction causes a trade-off between longevity and fertility. References have been cited in the revised manuscript according to your comment. However, since reduction of fecundity can be caused by a variety of factors, it cannot be a measure for lifespan extension.

8. Data related to Fig. 5: Fig 5B & 5C - Why are there different treatment time data for CI of *daf-16* but not *skn-1*? Fig. 5D: The data should also include N2 + SE or SA

Fig. 5B indicates the deterioration of sensory neurons in *daf-16* mutants over time. The data showed a significant decrease at 4-day-old. Based on the data, the CI of *skn-1* mutants was assayed at the 4-day-old time point.

Considering your suggestion, we performed additional experiments and the data is shown in the revised Fig. 5D along with the revised Supplementary Table 2.

9. The authors begin the discussion by stating on lines 183-185 - "Interestingly", FC made the nematodes resilient to infections and stressors, decreased AGEs levels, delayed senescence, and improved cognitive ability. I am not sure why this is "interesting" as this fact for various LAB or probiotics is well established.

Lines 197–205. Many papers have already showed the probiotic effects of lactic acid bacteria in *C. elegans* since our first report in 2007. In this manuscript, however, we could show the decreased AGEs and improved perception with enhanced host defense. This observation was interesting to us because increased AGEs and decreased perception of over aging were initially ameliorated by the human probiotic strain. This observation was also fascinating to us because it showed that the human probiotic

strain could ameliorate both increased AGEs and decreased perception of over aging in nematodes. This finding reinforces *C. elegans* as an alternative model to study effects of probiotics on senescence. Considering your comments, we have modified the description.

Reviewer #2 (Comments for the Author):

Review 2 Spectrum Komura et al

Komura et al. examined the effects of lactococci (LAB) on aging, lifespan and sensorimotor functions of *C. elegans*. They concluded that lactococci extend *C. elegans* life span likely through the Nrf2 transcription factor SKN-1. Cultivation of *C. elegans* on LAB slows down the accumulation of autofluorescent materials in the worm intestinal tissues, as well as delays the decline in chemotaxis functions. They provided some evidence that the exopolysaccharides (EPS) of LAB could play a role in some of the anti-aging effects of this probiotic microbe. The authors have invested significant efforts in a number of assays to show the beneficial effects of LAB on *C. elegans*, which should be acknowledged. However, the study is also complicated by a number of conceptual and technical concerns, as detailed below.

First of all, we would like to thank you for reviewing our manuscript. Considering the comments from both the reviewers, we have revised most of the figures based on the age of adulthood to avoid the confusion that arose in the previous version of the manuscript where both the day of treatment and the age were used. For clarity, we have now defined the age of young adult as nominal 0-day-old when worms reached the third day after hatching. All experiments were performed on 0-day-old adults with FC feeding. Consequently, all description of ages were revised throughout the manuscript.

Major comments:

1. The authors argue that it is unlikely that caloric restriction contributes to the pro-longevity effects of FC. However, several lines of evidence suggest that this possibility is real, including (1) FC-fed worms are smaller, and (2) FC-fed worms accumulate significantly less fat. The authors are recommended to study the accumulation of yolk protein in the treated hermaphrodites as a way to further evaluate the nutritional status of FC-fed worms. Citing intact brood size as a lack of caloric restriction is not appropriate, as this does not rule out caloric reduction (and the authors did not provide data of brood size, although described in the Methods section how this was performed). Or at least they should rephrase this part and leave the possibility open for further in-depth investigation.

Thank you for the constructive comments. Dietary restriction (DR) was used for the so-called calorie restriction (CR) in the case of *C. elegans* because it is hard to determine how many calories the worm can draw from bacteria. Moreover, CR and DR are believed to have a canonical effect on lifespan extension from yeast to mammals.

Lines 162–169 and 234–247. We agree that caloric restriction is a well-recognized intervention for anti-senescence and, as you pointed out, we cannot judge the adequate nutrition on the basis of the brood size alone (Fig. 4B). We have revised the description to suggest CR as one of the mechanisms by which FC extended lifespan of worms. However, we still think that some probiotic bacteria could extend the lifespan without CR because a mixture of OP50 and bifidobacteria, another lactic acid bacteria, prolonged the worms' lifespan (Komura et al. *Biogerontology* 2013; 14: 73-87). On the other hand, it is difficult to extend the lifespan of worms by CR while maintaining their fertility: it is common procedure to make the worms sterile with FuDR; otherwise, the worms would escape from the plate once DR starts. This suggests that some strains of probiotic bacteria produce factors that affect the longevity irrespective of CR.

The data on brood size had been described in the previous manuscript; this data is presented in line 166 and Fig. 4B in the current manuscript.

2. I do not support the use of cognitive function in this paper, as only chemotaxis to benzaldehyde was examined. *C. elegans* displays innate preference to benzaldehyde, and this is regarded as a sensory behavior but not cognition - cognition usually refers to more sophisticated behaviors such as learning and memory. To extrapolate the results of benzaldehyde chemotaxis to other behavioral aspects is not warranted unless more behavioral assays are conducted.

The title and lines 33, 97, 182, 199, and 269 are now edited. We agree with your explanation and have therefore, replaced “cognition” with “perception”.

3. The effects of EPS on longevity are questionable, since only one in three experiments showed slight but significant extension of life span compared to that of the control. The last sentence of the abstract (“both FC and EPS can affect longevity...”) should be revised. I need to further point out that the effects of EPS on life span are not examined in the *skn-1* mutant. Therefore, this statement in the abstract is unsupported and misleading.

Lines 55, 132–134, 191–192, and 225–227. Considering your comment, we have deleted the description that EPS may have beneficial effects.

4. Page 9, Line 174-177: Since EPS at a concentration of 10 mg/ml increases the expression of HO-1 to a level similar to that by heat-killed FC + 10 mg/ml EPS, it suggests that EPS at higher concentration could increase the expression of HO-1 without FC. Therefore, the conclusion that “it worked synergistically with heat-killed FC cells” is not supported by the data and must be revised. Or additional statistics should be provided to justify the authors' claim.

Lines 191–192. Considering your comment, we deleted the description that EPS may have beneficial effects in a synergistic manner.

5. Fig. 5A: The life span of the *daf-16* and *skn-1* mutants looks similar to that of the wild type, which is different from what most *C. elegans* studies had found. This is very strange. The authors should include N2 for experiments in Fig. 5A.

As you suggested, additional experiments were performed twice simultaneously to assay the lifespan and resilience to infection in not only mutant worms but also in N2 worms. The new data has been added to the respective figures and has also been included in the revised tables.

6. *skn-1(zu67)* is zygotically lethal, and one cannot have viable *zu67* homozygous adult worms for life span assays. The authors need to explain how life span experiments with *skn-1(zu67)* were performed.

We maintained the heterozygote that segregates Unc *skn-1*(zu67) heterozygotes, arrested eggs/larvae (nT1 homozygotes), and wild type *skn-1*(zu67) homozygotes. The *skn-1* homozygote itself can be non-lethal and grow by maternal effect, although it cannot produce offspring because of the absence of normal *skn-1*.

Minor points:

1. P5, Line 79: "gram positive" should be "Gram-positive". Please correct this wherever it applies.

CDC (<https://wwwnc.cdc.gov/eid/page/preferred-usage>) recommends that Gram should be capitalized and never hyphenated when used as Gram stain; gram negative and gram positive should be lowercase and only hyphenated when used as a unit modifier. However, we have revised according to your suggestion.

2. P6, Line 113: What does "ovary" in *C. elegans* mean?? This is confusing. There is something called uterus but not ovary in *C. elegans*.

Line 121. Thank you for your careful review. We have corrected it to "gonads".

3. P7, Line 129: "7 days of age" refers to worm age dated from newly hatched L1 larvae (= Day 4 adult)? But the standard in *C. elegans* aging research community refers to days in the adulthood as the age of the worm. This should be clearly indicated.

As mentioned earlier, to avoid confusion, we mentioned that day 1 of FC feeding referred to the third day after hatching: the 0-day-old worms were used for the experiments.

4. P7, Line 144-145: indicate "data not shown" or provide the data for cupric chloride or paraquat. Why are FC-fed worms more resistant to juglone but not to paraquat, since both induce oxidative stress?

Lines 157–159. FC failed to protect the worms from CuCl₂ and paraquat, and we did not show the data. In the first version of the manuscript, we mentioned "(data not shown)", but it was deleted in the 1st revision according to the comment of a previous reviewer. If the negative data are necessary, we can provide it in the supplementary figures. Alternatively, we could delete the sentence if the reviewer would agree. As of now, we cannot explain why FC was not effective against CuCl₂ and paraquat. However, it is not unusual that certain antioxidants protect only against specific kinds of oxidants. Since each oxidant and anti-oxidant seems to have its own distribution and bioavailability in vivo, a condition that is resistant to an oxidant cannot assure protection to all oxidants.

5. Page 9, Line 170: change "similar to" to "a homologue of the"

Line 186. Thank you. We have corrected it accordingly.

6. Page 11, Line 218: change "S. Enteritidis" to "enteritidis" (lower case italicized)

Line 31. Thank you for your suggestion. The systematic terminology has now been changed. *Salmonella enteritidis* should be indicated as *Salmonella enterica* subsp. *enterica* serovar. Enteritidis: *S. Enteritidis* is the accepted short name for this organism; however, for the sake of brevity, we have used the abbreviation SE for the pathogen in this revised manuscript.

7. Bar graphs should reveal all individual data points (Figs 1F, 2B, 3F, 3G, 4B, 5B, 5C, 6)

According to your recommendation, the figures were edited with new bar graphs.

8. Fig. 1A, 1B, 1G, 3A, 5A, 5D: the life span curves look weird with those thick vertical lines. In Fig. 1B, the vertical lines of the life span curves have multiple circles. Please correct these.

The data symbols were deleted from the graphs, and the graphs were indicated using simple lines.

9. Fig. 1C: Statistics? Incorporate A/B/C classes into the labels of locomotion groups.

To show the effects of FC on the health span of worms, the survival curves of the class-A worms were added to the additional Fig. 1D and evaluated using the log-rank test.

10. Fig. 1D: please show the error ranges (box plots with quartiles will be appropriate) in the data and also representative fluorescent images.

In the revised Fig. 1E, we have shown box plots. As we wrote to the editor last year, we have not presented the photograph because we have already published the data on the increased autofluorescence from AGEs of aged worms in reference no. 27 “npj Aging and Mechanisms of Disease” last year. Further, a colleague who provided photos with the fluorescence microscopy was not included in this project.

11. Fig. 1F: the unit at the Y-axis should be simplified.

The auxiliary unit is now indicated in a different manner.

12. Fig. 2: Please indicate benzaldehyde as the attractant in Figure 2 and the legend.

According to your recommendation, Figure 2 and its legend have been revised.

13. Fig. 5: please use the standard *C. elegans* nomenclature of mutant strains, including the allele name. Indicate in the B panel that this is the *daf-16* mutant, and in the C and D panels that it is the *skn-1* mutant. Also indicate the age of tested animals in Fig. 5C, not just in the legend.

We have added the information of the allele name and the age.

14. Fig. 6: add unit (mg/ml) to the EPS label at the X-axis.

We explained the amount of EPS used in the experiment in the figure legend to avoid long labels.

15. Remove redundancy of descriptions in the Figure legends. Some legends contain technical details that should be moved to the Methods section.

We have revised all figure legends to make them concise.

March 10, 2022

Prof. Yoshikazu Nishikawa
Tezukayamagakuin University
Faculty of Human Sciences
4-2-2 Harumidai
Sakai 5900113
Japan

Re: Spectrum00454-21R1 (Prolonged Lifespan, Improved Perception, and Enhanced Host Defense of *Caenorhabditis elegans* by *Lactococcus cremoris* sp. nov.)

Dear Prof. Yoshikazu Nishikawa:

Thank you for submitting your manuscript to Microbiology Spectrum. As you will see your paper is very close to acceptance. Please modify the manuscript along the lines I have recommended. As these revisions are quite minor, I expect that you should be able to turn in the revised paper in less than 30 days, if not sooner. If your manuscript was reviewed, you will find the reviewers' comments below.

When submitting the revised version of your paper, please provide (1) point-by-point responses to the issues I raised in your cover letter, and (2) a PDF file that indicates the changes from the original submission (by highlighting or underlining the changes) as file type "Marked Up Manuscript - For Review Only". Please use this link to submit your revised manuscript. Detailed instructions on submitting your revised paper are below.

Link Not Available

Sincerely,

Cheng-Yuan Kao

Reviewer comments:

Reviewer #1 (Comments for the Author):

Thank you for reviewing my comments and suggestions to improve the manuscript. There are still 1-2 issues that require further improvement.

Abstract: The addition of the first sentence is not sufficient to fully rationalise the aim of the study. In the authors' response, there is much more stated that indicates the novelty of this work when compared to other reports on probiotics testing in nematodes. Please also replace the term "living FC" (line 29) with "live FC". The term appears elsewhere too in the manuscript.

Lines 263-264 of the Conclusion section needs to be revised. The section begins by referring to the authors' earlier work but the subsequent sentences do not make it clear enough that they refer to the current work.

Reviewer #2 (Comments for the Author):

The authors satisfactorily addressed most of the points that I raised in the previous round of review. I have only a few minor suggestions on the writing of the manuscript, as detailed below.

Line 134: Change "chemotaxis intensity" to "chemotaxis index". Also indicate in Line 405 that CI means "chemotaxis index"; and Line 408, it is "chemotaxis assay", not "CI assay".

Line 170: "DAF-16..... is located at the end of the insulin..." consider changing this to "DAF-16..... is the target of regulation in the insulin....". Please make similar revision to where SKN-1 is mentioned (Line 172)

Preparing Revision Guidelines

- point-by-point responses to the issues I raised in your cover letter
- Upload a compare copy of the manuscript (without figures) as a "Marked-Up Manuscript" file.
- Each figure must be uploaded as a separate file, and any multipanel figures must be assembled into one file.
- Manuscript: A .DOC version of the revised manuscript
- Figures: Editable, high-resolution, individual figure files are required at revision, TIFF or EPS files are preferred

Please return the manuscript within 60 days; if you cannot complete the modification within this time period, please contact me. If you do not wish to modify the manuscript and prefer to submit it to another journal, please notify me of your decision immediately so that the manuscript may be formally withdrawn from consideration by Microbiology Spectrum.

Reviewer comments:

Reviewer #1 (Comments for the Author):

Thank you for reviewing my comments and suggestions to improve the manuscript. There are still 1-2 issues that require further improvement.

Abstract: The addition of the first sentence is not sufficient to fully rationalise the aim of the study. In the authors' response, there is much more stated that indicates the novelty of this work when compared to other reports on probiotics testing in nematodes. Please also replace the term "living FC" (line 29) with "live FC". The term appears elsewhere too in the manuscript.

Response: We thank you for your contributions to improving our manuscript. We have revised the manuscript, considering your suggestions.

Lines 25-29. We have provided more background information for the use of strain FC in *C. elegans* in the abstract. However, please note that the abstract is required to be no more than 250 words, and thus, we could not put more background information in the abstract.

We have corrected “living” to “live” in lines 29, 50, 68, 109, 110, 126, 219, 221, 224, and 487.

Lines 263-264 of the Conclusion section needs to be revised. The section begins by referring to the authors' earlier work but the subsequent sentences do not make it clear enough that they refer to the current work.

Response: Lines 264-270. In response to your suggestion, we have deleted the reference to our earlier work.

Reviewer #2 (Comments for the Author):

The authors satisfactorily addressed most of the points that I raised in the previous round of review. I have only a few minor suggestions on the writing of the manuscript, as detailed below.

Response: We wholeheartedly appreciate your support in improving our manuscript.

Line 134: Change "chemotaxis intensity" to "chemotaxis index". Also indicate in Line 405 that CI means "chemotaxis index"; and Line 408, it is "chemotaxis assay", not "CI assay".

Response: We have made the suggested changes in lines 133, 408, and 411.

Line 170: "DAF-16..... is located at the end of the insulin..." consider changing this to "DAF-16..... is the target of regulation in the insulin...". Please make similar revision to where SKN-1 is mentioned (Line 172)

Response: We have revised the indicated sentences according to your suggestion in lines 171 and 173.

April 8, 2022

Prof. Yoshikazu Nishikawa
Tezukayamagakuin University
Faculty of Human Sciences
4-2-2 Harumidai
Sakai 5900113
Japan

Re: Spectrum00454-21R2 (Prolonged Lifespan, Improved Perception, and Enhanced Host Defense of *Caenorhabditis elegans* by *Lactococcus cremoris* subsp. *cremoris*)

Dear Prof. Yoshikazu Nishikawa:

Your manuscript has been accepted, and I am forwarding it to the ASM Journals Department for publication. You will be notified when your proofs are ready to be viewed.

Sincerely,

Cheng-Yuan Kao
Editor, Microbiology Spectrum

Journals Department
Supplemental Material 1: Accept